# Improvement of Cognitive Function after Continuous Positive Airway Pressure Treatment for Subacute Stroke Patients with Obstructive Sleep Apnea: A Randomized Controlled Trial

**DOI:** 10.3390/brainsci9100252

**Published:** 2019-09-25

**Authors:** Howook Kim, Soobin Im, Jun il Park, Yeongwook Kim, Min Kyun Sohn, Sungju Jee

**Affiliations:** 1Department of Rehabilitation Medicine, Chungnam National University Hospital, Daejeon 35015, Korea; bulam5630@cnu.ac.kr (H.K.); ysbcool@cnuh.co.kr (S.I.); uniwater@naver.com (J.i.P.); kyu0922@cnuh.co.kr (Y.K.); mksohn@cnuh.co.kr (M.K.S.); 2Daejeon Chungcheong Regional Cardiocerebrovascular Center, Chungnam National University Hospital, Daejeon 35015, Korea; 3Daejeon Chungcheong Regional Medical Rehabilitation Center, Chungnam National University Hospital, Daejeon 35015, Korea

**Keywords:** subacute stroke, CPAP, obstructive sleep apnea, cognition

## Abstract

**Background:** Obstructive sleep apnea (OSA) is common after stroke. Various studies on continuous positive airway pressure (CPAP) therapy for OSA after stroke have been published. However, there have been no studies from Korea and Asia. The present Korean study aimed to determine whether CPAP treatment during inpatient rehabilitation of stroke patients with sleep disorders, especially OSA, improves function, cognition, sleep quality, and daytime sleepiness. **Methods:** This single-blind randomized controlled study included 40 stroke patients with OSA between November 2017 and November 2018. The patients were divided into the CPAP treatment group (CPAP and rehabilitation; *n* = 20) and control group (only rehabilitation; *n* = 20). The intervention period was 3 weeks. The primary outcomes were function and cognition improvements, and the secondary outcomes were sleep-related improvements. **Results:** CPAP treatment started at an average of 4.6 ± 2.8 days after admission. Both groups showed improvements in stroke severity, function, and cognition after the 3-week intervention. However, after the intervention, the degree of change in attention and calculation was significantly higher in the CPAP treatment group than in the control group. Additionally, the improvements in sleep quality and daytime sleepiness were greater in the CPAP treatment group than in the control group. **Conclusion:** CPAP treatment can improve cognitive function, sleep quality, and daytime sleepiness, and it should be considered as part of the rehabilitation program for patients with stroke. Our findings might help in the treatment of stroke patients with OSA in Korea.

## 1. Introduction

Approximately 795,000 people experience new strokes or recurrent strokes annually; 610,000 of these are first attacks, and 185,000 are recurrent attacks [1]. Stroke is a major cause of disability, and it impairs ambulation activities, activities of daily living (ADL), community integration, and quality of life (QOL), resulting in serious personal, social, and economic losses [1,2,3,4]. Because stroke recurrence is common, risk factors related to recurrence should be identified and controlled. Stroke has various risk factors [1], one of which is sleep disordered breathing (SDB) [1,5,6,7,8,9,10,11,12]. SDB is commonly observed after stroke, and it affects patients’ quality of sleep and QOL [13,14,15,16]. In addition, SDB affects cognition and functional recovery in patients with stroke [17,18,19]. According to Hermann et al., 60–70% of patients with stroke have SDB, which is much higher than that of the normal population [20]. Other studies have reported SDB in 50–70% of patients with acute and subacute stroke [21,22,23]. 

SDB in patients with stroke can lead to neurological deterioration and prolonged hospitalization [17,19,24,25], and it can affect post-stroke mortality [16,26,27,28] or short- and long-term prognoses in patients with stroke [14,18,29,30]. Obstructive sleep apnea (OSA) is the most common form of SDB, and it is caused by airflow disturbance due to airway obstruction [31,32,33]. OSA may improve in the subacute stages after stroke, but 50% of patients with stroke may still have an apnea–hypopnea index (AHI) >10 during the 2–3 months after onset [34]. Continuous positive airway pressure (CPAP) therapy is the main treatment for OSA [35,36]. In a study of SDB in patients with acute stroke, CPAP treatment improved patients’ exercise and function at 3 weeks after stroke onset, and compliance with treatment was high [29]. In addition, the AHI was significantly lower in the CPAP group than in the control group at 3 days after the stroke onset [37].

Parra et al. reported that the CPAP-treated group showed a significant improvement in the modified Rankin scale (mRS) score, Canadian Cardiovascular Society scale score, and cardiovascular disease and mortality rates compared to the control group [38]. According to a meta-analysis of randomized controlled trials (RCTs) published in 2018, 10 RCTs showed neurofunctional improvement with CPAP treatment [39], and improvement of long-term survival was confirmed in a study by Parra et al. [40]. Therefore, it is thought that the treatment of OSA after stroke has a good effect on the recovery of function after stroke. Studies on CPAP therapy for OSA after stroke have been continuously published [6,26,29,31,38,39,40,41]. In Korea and Asia, however, such studies have not been performed; thus, it is necessary to study this topic in order to improve the recovery of patients with stroke. Therefore, the purposes of this study were to determine whether CPAP treatment during inpatient rehabilitation of stroke patients with sleep disorders, especially OSA, significantly improves function or cognition, and to compare the improvement of sleep quality and daytime sleepiness scale score between patients with and without CPAP treatment.

## 2. Methods

### 2.1. Ethical Statement

Written consent was obtained from all patients or caregivers before study participation, and the study protocol was approved by the Institutional Review Board of Chungnam National University Hospital approved this study (No. 2017-08-051-002). The study was also registered in the International Clinical Trials Registry Platform database (CRiS, Clinical Research Information Service; Clinical Trial Registration No. KCT0003688).

### 2.2. Study Design

This study was performed in Chungnam University Hospital hospital, Republic of Korea using a single-blind RCT design. Staff at our hospital treated patients with hyperacute stroke in the cardiocerebrovascular center, and our hospital has a transfer system in the inpatient medical rehabilitation center during the acute and subacute periods after stroke. A blinded outcome analyzer collected cognitive and functional outcomes and sleep-related parameters. Previous studies have defined an AHI >30/h as severe OSA, but in this study, we evaluated patients with an AHI ≥20/h [42,43]. Patients with OSA were randomly assigned to either a CPAP treatment group or a control group (rehabilitation only). Clinical data were evaluated at the time of admission to the Department of Rehabilitation Medicine and after the 3-week intervention period.

### 2.3. Patients

This prospective study collected demographic and clinical data, including residual disability, activity limitations, and QOL, of patients with subacute stroke diagnosed by magnetic resonance imaging (MRI) or computed tomography (CT). Among the patients who were screened from November 2017 to November 2018 (*n* = 98), 43 patients were enrolled in this study. The data of the patients who participated in the final evaluation after the intervention (*n* = 40) were analyzed. The inclusion criteria were as follows: (1) A diagnosis of cerebral infarction or hemorrhage in the brain by CT or MRI; (2) only patients with predominant OSA (>50% of the respiratory events were obstructive type) exhibiting at least AHI ≥ 20/h were included, except those with central apnea or mixed apnea; (3) patients between 18 and 80 years of age; (4) patients admitted within 7 days to 6 months after stroke onset; (5) patients with cognitive functions capable of simple command obey; and (6) patients who provided informed consent. We excluded patients with any of the following: (1) A history of traumatic brain damage or brain tumor; (2) a diagnosis of mild-to-moderate obstructive sleep apnea (OSA); (3) baseline oxygen saturation <95%; (4) the presence of acute or chronic cardiopulmonary diseases that affects pulmonary function; (5) presence of neuromuscular diseases (e.g., amyotrophic lateral sclerosis and myasthenia gravis); and (6) an unstable medical condition preventing completion of the clinical trial.

### 2.4. Sleep Examination

We performed bedside sleep examination using a portable polysomnography called Stardust II (Philips Respironics Inc., Murrysville, PA, USA). This multichannel device recorded the following diagnostic parameters: Oxygen saturation, pulse rate, nasal airflow, and respiratory effort by chest wall motion. We performed the sleep examination from 9 PM to 6 AM, without overnight supervision. A trained sleep technologist analyzed the sleep data using the American Academy of Sleep Medicine criteria [44]. Hypopnea was defined as a reduction of airflow by ≥50% for at least 10 s, followed by oxygen desaturation ≥3%, and apnea was defined as a reduction of airflow by ≥90% for at least 10 s. Apneas with thoracic motion without chest wall motion or with an initial lack of motion followed by respiratory effort were classified as obstructive, central, or mixed, respectively. The AHI was defined as the mean number of apneas and hypopneas per hour. The obstructive apnea index and central apnea index were defined as the mean numbers of obstructive apneas and central apnea events per hour, respectively. The oxygen desaturation index (ODI) was defined as the mean number of oxygen desaturations ≥3% per hour. Sleep apnea was classified as obstructive or central, according to the type of predominant event. Predominant OSA was diagnosed when >50% of the respiratory events were of the obstructive type [38]. Sleep quality was assessed before and after the intervention by using the Epworth Sleepiness Scale (ESS) [45,46,47]. We obtained answers for the ESS from cooperative patients, or alternatively, from patients’ relatives. The ESS assesses daytime sleepiness, and it clinically defines a score >10 as excessive daytime sleepiness.

### 2.5. Randomization

After explaining the trial to the patients and receiving written informed consent, the physicians included in the study reported to the researcher that the patient was recruited into the study. The researcher allocated the patients to the CPAP treatment or control group using a random number table.

### 2.6. Intervention

Aside from nighttime CPAP therapy, all patients underwent conventional rehabilitation of the same degree on the schedule that was provided to them. Neither group performed treatments such as use of CNS stimulants such as methylphenidate, cognitive rehabilitation, or non-invasive brain stimulation. CPAP treatment was set up and monitored by one rehabilitation physician and two nurses who worked in the stroke rehabilitation ward. Personalized instructions were given to patients and caregivers before CPAP treatment, and written manuals were provided for the CPAP devices. Patients assigned to the CPAP treatment group were given a proper nasal or oronasal mask and continuous air pressure at night. Patients receiving CPAP treatment were treated with REMstar CPAP 60 Series A-Flex (Philips Respironics Inc., Murrysville, PA, USA) at a pressure setting with a 12 cmH2O. Patients were defined as having received adequate treatment if they had maintained CPAP for an average of >4 h over 3 weeks.

### 2.7. Demographic Data

#### 2.7.1. Primary Outcomes

Neurological and functional outcomes and sleep examination data were assessed before and after the intervention. The severity of stroke was assessed by the Korean version of the National Institute of Health Stroke Scale (NIHSS) and the functional outcome of patients by functional ambulation categories (FAC), modified Rankin Scale (MRS), and Berg Balance Scale (BBS). The activity of daily living was assessed by the Korean version of the modified Barthel index (K-MBI), and the quality of life was assessed by EuroQol-5 Dimension (EQ-5D). The patient’s cognitive status was evaluated by the Korean version of the Mini-Mental State Examination (K-MMSE). The improvement of neurological function and sleep quality was determined based on the difference between the results before and after the intervention. The primary outcome data analysis was performed by assessing the change of neurological severity and improvement of cognitive and functional assessments. We also assessed the level of improvement in each K-MMSE domain in the cognitive outcome. By using the K-MMSE data of each patient, we obtained data on “orientation to time, orientation to place, registration, attention and calculation, recall, language and drawing” [48], and evaluated the improvement level of data for each K-MMSE domain.

#### 2.7.2. Secondary Outcome

Secondary outcome data analysis was performed by evaluating improvement of the sleep examination data, such as obstructive apnea and AHI, and the improvement of the daytime sleepiness based on the ESS.

### 2.8. Statistical Analysis

Statistical analysis of data for patients who completed the intervention was performed, and the two-way analysis of variance was used to compare the differences in clinical data between the CPAP treatment and control groups. To evaluate the relationship between the additional CPAP treatment and each outcome, regression analyses were performed. Statistical analyses were performed using IBM SPSS Statistics version 23.0 (IBM Corp., Armonk, NY, USA). A *p*-value < 0.05 was considered statistically significant.

## 3. Results

We screened 98 patients with subacute stroke admitted to rehabilitation units between November 2017 and November 2018. Forty-three patients were predominant OSA patients with at least AHI ≥ 20. These patients were randomly assigned to the CPAP treatment group (*n* = 23) or control group (*n* = 20). However, 3 (13.0%) of 23 patients who started CPAP treatment refused CPAP because of mechanical discomfort. Therefore, the CPAP treatment group ultimately consisted of 20 patients, and control group included 20 patients (Figure 1). CPAP treatment started at an average of 4.6 ± 2.8 days after a patient with stroke was admitted to the Department of Rehabilitation Medicine. Patients were mainly men, and they had ischemic stroke and hypertension (Table 1). In the CPAP group, 2 patients with thrombolysis and 4 patients with thrombectomy were enrolled in the CPAP group. Three patients underwent hemorrhage removal and 3 patients underwent neurologic observation. In the control group, 2 patients with thrombolysis, 3 patients with thrombectomy, 2 patients with hemorrhage removal, and 2 patients with neurologic observation were enrolled in the control group. Most initial assessment data in both groups were similar. However, there was a difference between the CPAP treatment and control groups in the evaluation items related to quality of sleep, such as central apnea (5.5 ± 13.2 and 4.4 ± 14.0, respectively), obstructive apnea (26.5 ± 17.5 and 15.2 ± 13.4, respectively), mixed apnea (3.3 ± 6.9 and 5.2 ± 10.3, respectively), and AHI (44.4 ± 16.8 and 34.9 ± 17.2, respectively). The severity of stroke was similar in both groups (NIHSS score 6–7), and there was no significant difference in the measurement of functional or cognitive status. The subject’s body mass index (BMI) was less than 25 in both groups, which was less than other stroke subjects in the world. 

### 3.1. Primary Outcome Analysis (Functional and Cognitive Outcomes)

Both groups showed improvements in stroke severity, function, and cognition after the intervention compared with before the intervention (Table 2). The degree of improvement was similar in both groups, but the CPAP treatment group showed a better trend in stroke severity, balance and gait levels, and cognition than the control group. Although the degree of improvement of neurological and functional measures was better after the intervention in the CPAP treatment group than in the control group, no statistical significant difference was observed. However, the CPAP treatment group showed a significantly higher degree of change in the cognitive domain than the control group after the 3-week intervention period (K-MMSE score 4.0 ± 3.4 vs. 2.2 ± 1.9; *p* = 0.045). We assessed the cognitive-related domains of the patients to determine which one showed more improvement. Specifically, the CPAP treatment group showed significant improvement in the attention and calculation domain (*p* = 0.001), but no significant improvement in the other cognitive domains (Table 3).

### 3.2. Secondary Outcome Analysis (Sleep Examination Data)

CPAP treatment was performed for 3 weeks and follow-up evaluation was performed at 3 time points after treatment (Figure 2). All patients in the CPAP treatment group received CPAP treatment during the nighttime after the CPAP adaptation period, and they were highly compliant (>4 h/day, ≥5 days a week). However, in this study, compliance with CPAP treatment was not accurately measured. During the hospital stay, the nurse in the ward checked the patients’ application of the CPAP machine and educated the caregiver on how to measure the wearing time. As a result of the sleep examination, the CPAP treatment group showed better improvement in the ESS score, central apnea, obstructive apnea, AHI, snore flag index, and ODI than the control group. Especially, the CPAP treatment group (*n* = 20) showed a significant decrease in the AHI (17.9 ± 12.8 vs. −3.0 ± 9.7, *p* = 0.001) and obstructive apnea (−13.0 ± 14.1 vs. 1.6 ± 10.6, *p* = 0.001) compared with the control group. In addition, significant improvement was observed in the CPAP treatment group compared to the control group in the ESS score (−2.3 ± 2.3 vs. 0.6 ± 3.3, *p* = 0.003) (Table 4). There was no significant reduction in mixed apnea in the CPAP treatment and control groups (−0.5 ± 7.1 and −0.8 ± 11.3, respectively; *p* = 0.715). Additionally, to confirm the correlation between improvement of the AHI by CPAP treatment and cognitive improvement, the change of the K-MMSE score according to the improvement of AHI was evaluated. In the regression analysis, ΔK-MMSE (*ß* = 0.071, 95% confidence interval [CI], 0.068–0.176, *p* = 0.033) and ΔESS (*ß* = 0.109, 95% CI, 0.096–0.257, *p* = 0.002) were significantly correlated with ΔAHI (Table 5).

## 4. Discussion

To the best of our knowledge, this is the first RCT to evaluate sleep quality and cognitive and functional statuses in Korean patients with subacute stroke and OSA after CPAP treatment more than 3 weeks. We found that CPAP therapy can help improve cognitive function such as sleep quality and sleepiness, as well as attention and calculation. Some of the main outcomes of the CPAP group, such as cognitive outcome, severity of sleep apnea, and quality of sleep, improved, but improvement in most assessments was not statistically significant compared to the control group. In contrast to a previous study [29], we did not find significant improvements in the CPAP treatment group in functional outcomes, including neurological status and ADL. Because the CPAP treatment group and control group showed similar improvements in functional areas compared to before the intervention, the role of rehabilitation therapy in the functional area may be judged to be more important than sleep apnea treatment. However, since CPAP therapy was only performed during a short time, there may be a limit to assessing the effectiveness of CPAP therapy in relation to patients’ functional outcome. In addition, as the CPAP and rehabilitation group and CPAP-only group were not compared directly, the superiority of each treatment could not be determined.

Previous studies have shown an improvement in cognitive function after OSA and stroke with CPAP treatment, which is consistent with the results seen in patients of the CPAP treatment group of the present study. We used the K-MMSE to evaluate cognitive function improvement and found significant improvement in the attention and calculation domain [49,50]. Other studies have shown that stroke patients with OSA have greater impairment in cognitive function, including attention and executive function, than stroke patients without OSA [14]. We also compared daytime sleepiness between the CPAP treatment and control groups, and found a better level of improvement in the CPAP treatment group than in the control group. These results may have a positive impact on patients’ cognitive and functional improvement by inducing greater participation in the rehabilitation treatment program. In this study, only 3 of the 23 patients in the CPAP group with OSA refused treatment. The remaining patients were treated with CPAP for more than 4 hours a day for >3 weeks because all the patients were hospitalized, and various medical professionals, such as doctors and nurses, who were trained in CPAP treatment checked the patient’s condition at night and encouraged the use of CPAP. Additionally, because SDB was diagnosed early, CPAP treatment could be applied early, which could have resulted in the higher compliance with CPAP treatment. 

A recent meta-analysis of CPAP treatment for sleep apnea in patients with stroke noted that sleep apnea after stroke may affect perfusion and oxygenation of the penumbra, which may adversely affect neural damage and stroke outcome [39,51]. The early application of CPAP may prolong survival of the penumbra, resulting in clinical and imaging improvement in patients after stroke. Furthermore, the application of ongoing CPAP therapy can independently contribute to improving cognitive impairment, drowsiness, and depression, leading to better participation of the patient in the rehabilitation program, which may have a more positive impact on recovery after stroke. Evidence for the beneficial effects associated with neurological, cognitive, or long-term survival of CPAP after stroke have not yet been clarified, but further RCTs and the present study may support a positive effect of CPAP therapy, as it may be an additional treatment option for cognition and function improvement in patients with stroke. 

This study has several limitations that remain to be addressed. First, the sample size of the study was small. Of the 98 patients who underwent sleep examination, 43 were diagnosed with OSA and only 40, except 3 patients who were withdrawn from the study because of medical condition deterioration or refusal to CPAP treatment, participated in this RCT. The prevalence of sleep apnea in this study was lower than that in previous studies (71%) [52] because we used strict diagnostic criteria for OSA. Additionally, the use of portable polysomnography also tends to underestimate AHI. Portable devices without electroencephalography recording capability cannot distinguish between awake and sleep states. Therefore, it is possible that the number of patients who could participate in this study was limited by this factor. Second, because of the limited hospitalization period, CPAP treatment was performed during a short intervention period. To identify the effectiveness of CPAP treatment, a 4-week or longer treatment period may be needed. Further, after the intervention period, the control group was unable to perform CPAP treatment and did not perform long-term follow-up after discharge to further evaluate the effect of CPAP treatment on OSA. Therefore, when interpreting the results of this study, the characteristics of the hospital’s patients and the health policy situation in Korea should be considered. Third, CPAP compliance may have increased because of treatment encouragement by doctors and nurses rather than by family and caregivers. Since the CPAP machine itself is unable to accurately monitor patients’ compliance, the accuracy of CPAP compliance cannot be confirmed. Although continuing CPAP therapy may be beneficial to patients after discharge, there may be a limit to the maintenance of CPAP treatment if the family or caregivers do not receive awareness or training for CPAP treatment. Maintaining high CPAP compliance over a prolonged period may be more beneficial for the patient. Fourth, patients did not undergo a sleep examination before the diagnosis of stroke. Therefore, it was not possible to determine whether they already had sleep apnea. Previous studies have suggested that sleep apnea is more likely to precede the onset of stroke and that sleep apnea improves in most patients after the acute phase of stroke [18,52]. Fifth, patients with severe neurological deficits showed more severe sleep apnea but no significant functional improvement in CPAP treatment. This study compared the mean values of patients with various severities of stroke, and consequently, the severity of stroke may directly affect the efficacy of CPAP treatment. Lastly, neurostimulants such as methylphenidate have not been used, but we did not evaluate sleep-related medications (sedatives) in subacute stroke patients with OSA. Therefore, aside from using CPAP, we could not confirm the change of sleep quality or weekly daytime sleepiness caused by sleep-related medication.

## 5. Conclusions

Sleep apnea is a common disease in patients with subacute stroke, and it can aggravate neurological and functional statuses. Clinicians should assess patients’ sleep status and adequately treat sleep disturbances during the rehabilitation of subacute stroke. The beneficial effects of CPAP treatment found in this study suggest that this treatment should be considered as part of the rehabilitation program for patients with stroke. The treatment of choice for OSA is CPAP treatment. When applied to subacute stroke patients with OSA for a short period, there was improvement in sleep quality, daytime sleepiness, and cognitive function. Patients with high compliance and long-term CPAP treatment may have a greater benefit. Appropriate CPAP treatment can improve patients’ overall function by improving their cognitive function and daytime sleepiness, and consequently, inducing their participation in a rehabilitation program. Further research is needed on the improvements in neurological and functional statuses among stroke patients who have received long-term CPAP treatment.

## Figures and Tables

**Figure 1 brainsci-09-00252-f001:**
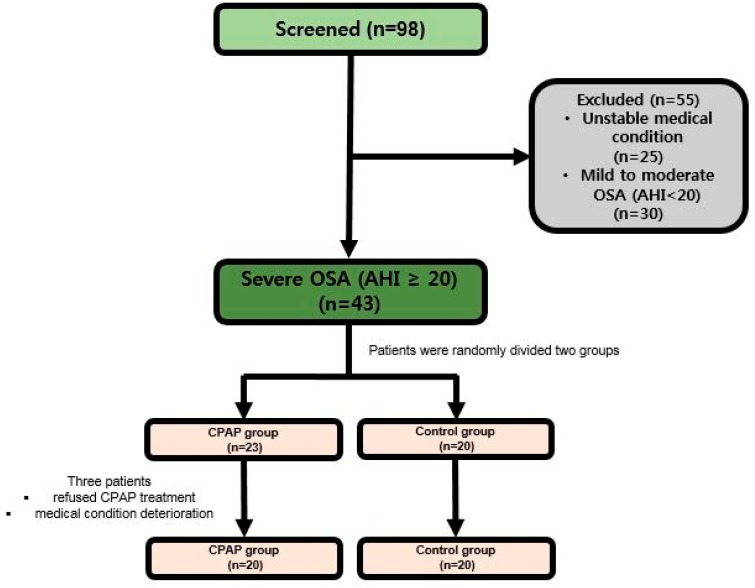
Flow chart to study population. PSG, polysomnography; OSA, obstructive sleep apnea; CPAP, continuous positive airway pressure.

**Figure 2 brainsci-09-00252-f002:**
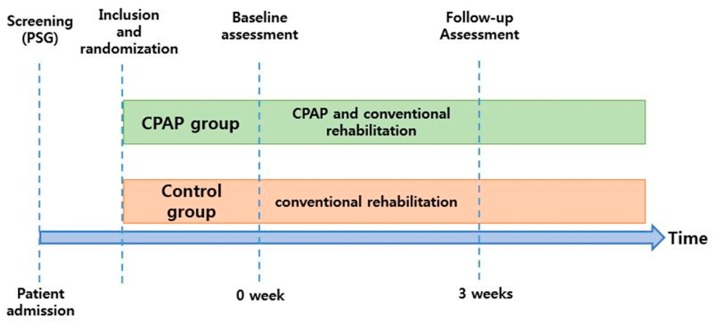
Study design. PSG, Polysomnography.

**Table 1 brainsci-09-00252-t001:** Clinical characteristics of intervention group.

	CPAP (*n* = 20)	Control (*n* = 20)	*p*-Value
Sex, *n* (men/women)	13/7 (65.0/35.0%)	16/4 (80.0/20.0%)	0.429
Age (years)	63.3 ± 13.1	66.9 ± 12.3	0.369
Type of stroke, *n* (ischemic/hemorrhagic)	14/6 (70.0/30.0%)	16/4 (80.0/20.0%)	0.602
Lesion type, *n* (Supratentorial/Infratentorial)	14/6 (70.0/30.0%)	11/9 (55.0/45.0%)	0.429
Ischemic group primary treatment, *n* (Thrombolysis/Thrombectomy)	2/4 (10.0/20.0%)	2/3 (10.0/15.0%)	0.000
Hemorrhagic group primary treatment, *n* (observation/hemorrhage removal)	3/3 (50.0/50.0%)	2/2 (50.0/50.0%)	0.000
BMI	23.3 ± 3.7	24.4 ± 3.9	0.370
LOS	38.6 ± 11.4	37.8 ± 14.8	0.849
NIHSS	6.7 ± 3.5	6.5 ± 5.7	0.869
K-MMSE	18.6 ± 7.7	17.5 ± 9.1	0.668
FAC	1.3 ± 1.5	1.7 ± 2.0	0.474
MRS	3.9 ± 1.0	3.5 ± 1.2	0.207
BBS	15.5 ± 17.1	22.9 ± 22.9	0.251
K-MBI	43.1 ± 26.5	45.8 ± 31.6	0.775
HTN (+/−)	13/7 (65.0/35.0%)	15/5 (75.0/25.0%)	0.602
DM (+/−)	4/16 (20.0/80.0%)	7/13 (35.0/65.0%)	0.429
EQ-5D	0.3 ± 0.3	0.3 ± 1.3	0.612
ESS	6.0 ± 5.4	6.7 ± 5.2	0.677
Central apnea	5.5 ± 13.2	4.4 ± 14.0	0.796
Obstructive apnea	26.5 ± 17.5	15.2 ±13.4	0.028 *
Mixed apnea	3.3 ± 6.9	5.2 ± 10.3	0.503
Hypopnea	9.1 ± 8.9	10.1 ± 7.4	0.699
AHI	44.4 ± 16.8	34.9 ± 17.2	0.085
Snore flag index	62.6 ± 76.3	42.0 ± 63.9	0.359
Desaturation index	43.3 ± 18.4	34.4 ± 20.2	0.156

OSA, obstructive sleep apnea; BMI, body mass index; LOS, length of stay; NIHSS, Korean version of the National Institute of Health Stroke Scale; MMSE, Korean version of the Mini-Mental State Examination; FAC, functional ambulation categories; MRS, modified Rankin Scale; BBS, Berg Balance Scale; K-MBI, Korean version of the modified Barthel index; HTN, hypertension; DM, diabetes mellitus; EQ-5D, EuroQol-5 Dimension; ESS, Epworth Sleepiness Scale; AHI, apnea–hypopnea index; * *p*-value < 0.05.

**Table 2 brainsci-09-00252-t002:** Comparison of clinical outcome between CPAP and control group.

	CPAP (*n* = 20)	Control (*n* = 20)	*p*-Value
ΔNIHSS	−1.5 ± 1.3	−1.1 ± 1.5	0.157
ΔMMSE	4.0 ± 3.4	2.2 ± 1.9	0.045 *
ΔFAC	0.8 ± 1.0	0.9 ± 1.0	0.862
ΔMRS	−0.8 ± 0.8	−0.4 ± 0.6	0.142
ΔBBS	10.0 ± 10.3	8.7 ± 10.7	0.583
ΔK-MBI	14.0 ± 9.8	13.5 ± 9.9	0.873
ΔEQ-5D	0.2 ± 0.2	0.2 ± 0.3	0.282

Δ, difference in score between pre-intervention–post-intervention. CPAP, continuous positive airway pressure; NIHSS, Korean version of the National Institute of Health Stroke Scale; MMSE, Korean version of the Mini-Mental State Examination; FAC, functional ambulation categories; MRS, modified Rankin Scale; BBS, Berg Balance Scale; K-MBI, Korean version of the modified Barthel index; EQ-5D, EuroQol-5 Dimension. * *p*-value < 0.05.

**Table 3 brainsci-09-00252-t003:** Comparison of cognitive outcome between CPAP and control group.

	CPAP (*n* = 20)	Control (*n* = 20)	*p*-Value
ΔOrientation to time (5)	0.7 ± 1.0	0.1 ± 1.4	0.055
ΔOrientation to place (5)	1.0 ± 1.1	0.6 ± 0.9	0.155
ΔRegistration (3)	0.1 ± 0.4	0.2 ± 0.7	0.554
ΔAttention and calculation (5)	1.4 ± 0.8	0.3 ± 1.3	0.001 *
ΔRecall (3)	0.3 ± 1.1	0.4 ± 0.6	0.558
ΔLanguage (8)	0.5 ± 0.9	0.8 ± 1.1	0.501
ΔDrawing (1)	0.1 ± 0.4	0.1 ± 0.6	0.710
ΔTotal (30)	4.0 ± 3.4	2.2 ± 1.9	0.045 *

Δ, difference in score between pre-intervention–post-intervention. CPAP, continuous positive airway pressure. * *p*-value < 0.05.

**Table 4 brainsci-09-00252-t004:** Comparison of daytime sleepiness index and polysomnographic data between CPAP and control group.

	CPAP (*n* = 20)	Control (*n* = 20)	*p*-Value
ΔESS	−2.3 ± 2.3	0.6 ± 3.3	0.003 *
ΔCentral apnea	−2.7 ± 10.2	−0.2 ± 2.9	0.449
ΔObstructive apnea	−13.0 ± 14.1	1.6 ± 10.6	0.001 *
ΔMixed apnea	−0.5 ± 7.1	−0.8 ± 11.3	0.715
ΔHypopnea	−1.8 ± 8.1	−3.7 ± 5.0	0.378
ΔAHI	−17.9 ± 12.8	−3.0 ± 9.7	0.001 *
ΔSnore flag index	−23.5 ± 54.8	0.7 ± 70.3	0.441
ΔDesaturation index	−16.2 ± 14.9	−7.7 ± 19.6	0.133

Δ, difference in score between pre-intervention and post-intervention. CPAP, continuous positive airway pressure; ESS, Epworth Sleepiness Scale; AHI, apnea–hypopnea index. * *p*-value < 0.05.

**Table 5 brainsci-09-00252-t005:** Regression analysis concerning CPAP treatment effectiveness.

	*β*	Adjusted R^2^	*p*-Value
ΔAHI			
ΔMMSE	0.071	0.114	0.033 *
ΔESS	0.109	0.220	0.002 *

Δ, difference score between pre-intervention and post-intervention. AHI, apnea–hypopnea index; MMSE, Korean version of the Mini-Mental State Examination; ESS, Epworth Sleepiness Scale. * *p*-value < 0.05.

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
