# Peer review of "Improvement of Cognitive Function after Continuous Positive Airway Pressure Treatment for Subacute Stroke Patients with Obstructive Sleep Apnea: A Randomized Controlled Trial"

_brainsci, 2019, doi:10.3390/brainsci9100252_

Round 1

Reviewer 1 Report

This study sought to determine whether CPAP treatment during inpatient rehabilitation of stroke patients with sleep disorders, especially OSA, improves function, cognition, sleep quality, and daytime sleepiness, and to compare the improvement in sleep quality and daytime sleepiness scale score between patients with and without CPAP treatment. To do this, the authors examined 40 stroke patients with OSA and provided them with CPAP treatment and rehabilitation (CPAP treatment group) or only rehabilitation (control group) for 3 weeks. It was found that both groups showed improvements in stroke severity, function, and cognition after 3 weeks. However, the CPAP treatment group showed significant improvements in the degree of change in attention and calculation as compared to the control group suggesting that CPAP treatment can improve patients’ overall function by improving their cognitive function and daytime sleepiness, and consequently, inducing their participation in a rehabilitation program.

Overall, this is a well-described study. The manuscript is also well written. The hypothesis is clear and the results are fairly conclusive. I approve this manuscript. Errors in typographical or formatting must be corrected.

Author Response

We appreciated the reviewer’s comment.

We greatly appreciate the constructive criticisms from reviewer 1.

Reviewer 2 Report

The paper by dr. Howook and colleagues is very interesting, however I have some comments.

Introduction:

Is data from the first sentence in the introduction for all strokes or only for ischemic stroke? 

The reference for "stroke has various..." should be added (e.g. Ann Clin. Biochem. 2017;54(3):378-85).

The introduction is too long and should be condensed.

Methods and results:

Why only patients with AHI ≥ 20/h were included?

Please add unit for AHI in inclusion criteria.

Overall, data on patients' treatment is missing. How many patients were treated by using iv thrombolysis or trombectomy?

Is there any possibility to show commonly known risk factors for strokes in table 1? The authors showed only DM.

The authors described hypertension in the text but they did not show it in table 1.

It would be very interesting if the authors could present similar analyses only for ischemic stroke patients. 

Overall, the paper is well-written, the methodology and the finding are both clearly described.

Author Response

We greatly appreciate the constructive criticisms from reviewer 2.

Below are our responses to their comments.

1. Is data from the first sentence in the introduction for all strokes or only for ischemic stroke?

: I appreciated the reviewer’s comment. Both ischemic and hemorrhagic strokes are covered.

2. The reference for "stroke has various..." should be added (e.g. Ann Clin. Biochem. 2017;54(3):378-85).

:I have checked your changes and added a reference.

3. The introduction is too long and should be condensed.

: I appreciated the reviewer’s comment. I tried to summarize as much as possible, but it seems longer because I wanted to mention the importance of OSA, which is considered a risk factor for stroke, and I wanted to explain the need for OSA management.

4. Why only patients with AHI ≥ 20/h were included?

: The aim of this study was to investigate patients with subacute stroke who had moderate or higher OSA severity. Other literature, including reference no.14, found moderate to severe OSA when AHI was above 15. This study was performed on a subacute stroke patient. In the reference no.38 studies involving acute stroke patients, the study was performed on patients with AHI of 20 or more. Therefore, AHI was set to 20 to see the CPAP effect of the subacute patient group compared to acute stroke patients.

5. Please add unit for AHI in inclusion criteria.

: I appreciated the reviewer’s comment. I checked the changes and added that.

6. Overall, data on patients' treatment is missing. How many patients were treated by using iv thrombolysis or trombectomy?

: I appreciated the reviewer’s comment. I checked the changes and added that.

: In the CPAP group, four patients with thrombectomy, one patient with thrombolysis, six patients with antiplatelet, two patients with Non-Vitamin K Antagonist Oral Anticoagulants, and one patient with warfarin were enrolled in the CPAP group. Two patients underwent hemorrhage removal and four patients underwent neurologic observation. In the control group, 3 patients were treated with Thrombectomy, 11 patients with antiplatelet, 2 patients with warfarin, 2 patients with hemorrhage removal and 2 patients with neurologic observation.

7. Is there any possibility to show commonly known risk factors for strokes in table 1? The authors showed only DM. The authors described hypertension in the text but they did not show it in table 1.

: Among the risk factors of stroke, this study confirmed only hypertension and diabetes. Table 1 shows the data for hypertension.

8. It would be very interesting if the authors could present similar analyses only for ischemic stroke patients.

: We appreciated the reviewer’s comment. We will reflect this in a follow-up study.

Reviewer 3 Report

This study looks at the effect of CPAP in Korean patients who have suffered a stroke.  It found CPAP was associated with an improvement in the Korean MMSE particularly the attention and calculation scores but no change in .

The authors are to be commended on undertaking this first in Korea study but there is much missing from this paper and I do not think it can be published in its current state.

There is no discussion of the power of this study to detect differences between the treatment and non treated groups. There was no sham treatment and although the data was collected by an outcome analyser it is not possible to avoid the sharing of treatment information Patients were eligible for CPAP from 7 days to 6 months after stroke onset.  There is no data on the time since stroke onset. 6 months is well outside the normal post stroke period where the particular events of a stroke might be amenable to CPAP How was a treatment pressure of 12cm decided.  Epworth Sleepiness score was used.  This includes 2 questions pertaining to driving or being a passenger in a car.  How was that adjusted for in the 3 week post intervention assessment when subjects were still in hospital The authors discuss an "improvement in AHI" between the treatment and control groups.  What does this mean.  Did the control group undergo a second study?  Was the treatment groups AHI taken from the CPAP device? The pre treatment mean AHI was 44. The ∆AHI was 17.9 - what was the mean on treatment AHI? In terms of the above results many patients would not have had a sufficient reduction in AHI See item 4 above Why was the average nightly use of CPAP not read from the machine?  Asking family to document how long the patient used the device seems inadequate.  I assume Korean families sleep with the subjects but presumably were not awake all night. The BMI of the subjects was under 25 - much lighter than most stroke subjects elsewhere in the world.  The authors might comment on this

Most of the above issues would limit the effect of the study and reduce efficacy of the intervention - the lack of a sham treatment and lack of blinding would increase the chance of an erroneous positive cnoclusion

Author Response

We greatly appreciate the constructive criticisms from reviewer 3. Below are our responses to their comments.

1. There is no discussion of the power of this study to detect differences between the treatment and non treated groups. 

: I appreciated the reviewer’s comment. This study is about the effect of CPAP treatment on patients with OSA in subacute stroke patients. There was a difference in the improvement of sleep quality between treatment and non-treatment groups. There was also a difference in the degree of cognition improvement (especially attention and calculation domain) between the treatment and non-treatment groups. Appropriate CPAP treatment will improve the sleep quality of the patient and improve the attention of the patient, which may result in improvement in the patient's participation in rehabilitation.

2. There was no sham treatment and although the data was collected by an outcome analyser it is not possible to avoid the sharing of treatment information Patients were eligible for CPAP from 7 days to 6 months after stroke onset. 

: We agree with the reviewer’s opinion. We are sorry for the lack of the Sham Group. I will refer to the follow-up study later. However, there should be no possibility of sharing treatment information by Outcome analyser. CPAP treatment was performed by one physician and two nurses at stroke rehabilitation ward. None of the medical personnel participated as an outcome analyzer.

3. There is no data on the time since stroke onset. 6 months is well outside the normal post stroke period where the particular events of a stroke might be amenable to CPAP How was a treatment pressure of 12cm decided. 

: Previous studies (reference no. 29, 38, 39, 50, 51, etc.) do not suggest a post stroke period for applying CPAP. So we did not collect data specifically about time after stroke. In addition, in most studies, the treatment pressure was varied within the range of 6 ~ 14cmH2O. Therefore, in this journal, 12cmH2O was set to provide a similar level of treatment as the previous studies.

4. Epworth Sleepiness score was used.  This includes 2 questions pertaining to driving or being a passenger in a car.  How was that adjusted for in the 3 week post intervention assessment when subjects were still in hospital. 

: I appreciated the reviewer’s comment. The ESS evaluation was also performed in reference no.38, and the patient's daytime sleepiness scale was also used. In this study, the same evaluation tool was adopted. The assessment was conducted twice before and after intervention during hospitalization. Rather than comparing the absolute values of each data, the evaluation was focused on the differences before and after the intervention. Questions related to driving a car or passengers were rated 0 on both initial and secondary assessments.

5. The authors discuss an "improvement in AHI" between the treatment and control groups.  What does this mean.  Did the control group undergo a second study? 

: I appreciated the reviewer’s comment. In this study, we looked mainly at changes (delta values) of sleep-related data as outcomes. The absolute value of AHI was improved in both groups, but the degree of change in AHI was different in both groups, and the difference was greater in the CPAP treatment group. Both groups performed up to a second assessment because the extent of the change had to be confirmed.

6. Was the treatment groups AHI taken from the CPAP device?

: I appreciated the reviewer’s comment. All patients' AHI was measured by portable polysomnography, not by CPAP devices. Of the patients diagnosed with moderate to severe OSA by portable polysomnography, the sleep apnea treatment was performed using the CPAP device only in the treatment group.

7. The pre treatment mean AHI was 44. The ∆AHI was 17.9 - what was the mean on treatment AHI?

: I appreciated the reviewer’s comment. The delta AHI value is an indicator of how well the absolute value of AHI has improved after treatment.

8. In terms of the above results many patients would not have had a sufficient reduction in AHI See item 4 above. 

: I appreciated the reviewer’s comment. The data shown in Table 4 lists all the data that can be confirmed by the polysomnography. The subjects in this study are those with subacute stroke who have obstructive sleep apnea, and therefore are more meaningful for data such as Obstructive Apnea and AHI that meet the diagnostic criteria of OSA patients.

9. Why was the average nightly use of CPAP not read from the machine?

: I appreciated the reviewer’s comment. Previous studies did not specifically analyze average nighttime usage. In addition, the CPAP device used in this study did not have an average night-use analysis function, so we did not analyze the data. I will refer to it for further study.

10. Asking family to document how long the patient used the device seems inadequate. I assume Korean families sleep with the subjects but presumably were not awake all night.

: I appreciated the reviewer’s comment. In this study, two nurses and a caregiver at night were used to encourage patients to improve CPAP compliance and to verify that CPAP devices were worn correctly. CPAP usage hours were not specifically requested to be documented.

11.The BMI of the subjects was under 25 - much lighter than most stroke subjects elsewhere in the world.  The authors might comment on this

: I appreciated the reviewer’s comment. I checked the changes and added that.

12. Most of the above issues would limit the effect of the study and reduce efficacy of the intervention - the lack of a sham treatment and lack of blinding would increase the chance of an erroneous positive conclusion.

: We totally agree with the reviewer’s opinion. As I've answered many of the comments above, the absence of the sham group is a sad. But, there is no lack of blinding, so the chances of the chance of an erroneous positive conclusion are not high.

Round 2

Reviewer 2 Report

In fact, the authors did not respond to the different aspects which were mentioned in the first review of the manuscript. But more importantly, taking into consideration that different groups in terms of treatment modalities were analysed, the study is inconclusive. The patients included in the study were heterogeneous with regard to treatment and cannot be analysed as a single population.

Author Response

In fact, the authors did not respond to the different aspects which were mentioned in the first review of the manuscript. But more importantly, taking into consideration that different groups in terms of treatment modalities were analysed, the study is inconclusive. The patients included in the study were heterogeneous with regard to treatment and cannot be analysed as a single population.

: I apologize for not having enough first answer to comments.

The patient's data was examined again precisely to confirm the treatment distribution and added to Table 1. Not all of the patients with ischemic stroke received primary treatment, such as thrombolysis or thrombectomy.
The distribution of primary treatment for ischemic stroke and the distribution of hemorrhagic stroke between CPAP and control groups showed similar trends, indicating that the population is not heterogenous.

 The aim of this study was not to analyze data for patients with ischemic stroke, but to report the outcome of CPAP treatment in patients with subacute stroke, including both ischemic and hemorrhagic stroke. We did not intend to conduct research on patients who had exactly the same neurological treatment. To compare the severity between individual patients, we used NIHSS as mentioned in Table 1, and both groups showed similar NIHSS.

Please review NIHSS if your reviewer's comment is on the primary treatment of thrombolysis or thrombectomy to identify areas of severity of stroke.

Reviewer 3 Report

No further comments

Author Response

We appreciated the reviewer’s comment.

We greatly appreciate the constructive criticisms from reviewer 3.